# Elevated CSF GAP-43 is associated with accelerated tau accumulation and spread in Alzheimer's disease

Nicolai Franzmeier [1,2,3] ✉, Amir Dehsarvi [1], Anna Steward[1], Davina Biel[1], Anna Dewenter [1], Sebastian Niclas Roemer [1], Fabian Wagner[1], Mattes Groß[1,4], Matthias Brendel [2,4], Alexis Moscoso [3], Prithvi Arunachalam[3], Kaj Blennow[3,5], Henrik Zetterberg [3,5,6,7,8,9], Michael Ewers [1,10] & Michael Schöll [3,11,12]

In Alzheimer's disease, amyloid-beta (Aβ) triggers the trans-synaptic spread of tau pathology, and aberrant synaptic activity has been shown to promote tau spreading. Aβ induces aberrant synaptic activity, manifesting in increases in the presynaptic growth-associated protein 43 (GAP-43), which is closely involved in synaptic activity and plasticity. We therefore tested whether Aβ-related GAP-43 increases, as a marker of synaptic changes, drive tau spreading in 93 patients across the aging and Alzheimer's spectrum with available CSF GAP-43, amyloid-PET and longitudinal tau-PET assessments. We found that (1) higher GAP-43 was associated with faster Aβ-related tau accumulation, specifically in brain regions connected closest to subject-specific tau epicenters and (2) that higher GAP-43 strengthened the association between Aβ and connectivity-associated tau spread. This suggests that GAP-43-related synaptic changes are linked to faster Aβ-related tau spread across connected regions and that synapses could be key targets for preventing tau spreading in Alzheimer's disease.

In Alzheimer's disease (AD), the accumulation of cerebral amyloid-beta (Aβ) plaque pathology is assumed to initiate a cascade of pathological processes, including the expansion of hyperphosphorylated tau pathology from the temporal lobe to the cortex, thereby driving neurodegeneration and cognitive decline[1,2]. From a mechanistic point of view, in vitro and animal studies suggest that tau pathology spreads across interconnected neurons, most likely through synapses[3,4]. In line with these findings, we and others could previously confirm that the expansion of tau pathology in AD patients specifically follows the connectivity pattern of tau epicenters in which tau pathology emerges first[5-8], suggesting that neuronal connections and synapses are the putative pathways along which tau spreads in AD.

Synaptic activity has been shown to play a crucial role in the trans-synaptic spreading of tau, where higher synaptic activity and connectivity are associated with accelerated secretion of hyperphosphorylated tau at the synapse and faster trans-synaptic tau spread[3].

[1]Institute for Stroke and Dementia Research (ISD), University Hospital, LMU Munich, Munich, Germany. [2]Munich Cluster for Systems Neurology (SyNergy), Munich, Germany. [3]University of Gothenburg, The Sahlgrenska Academy, Institute of Neuroscience and Physiology, Department of Psychiatry and Neurochemistry, Mölndal and Gothenburg, Sweden. [4]Department of Nuclear Medicine, University Hospital, LMU Munich, Munich, Germany. [5]Clinical Neurochemistry Laboratory, Sahlgrenska University Hospital, Mölndal, Sweden. [6]Department of Neurodegenerative Disease, UCL Institute of Neurology, Queen Square, London, UK. [7]UK Dementia Research Institute at UCL, London, UK. [8]Hong Kong Center for Neurodegenerative Diseases, Clear Water Bay, Hong Kong, China. [9]Wisconsin Alzheimer's Disease Research Center, University of Wisconsin School of Medicine and Public Health, University of Wisconsin-Madison, Madison, WI, USA. [10]German Center for Neurodegenerative Diseases (DZNE), Munich, Germany. [11]Wallenberg Centre for Molecular and Translational Medicine, University of Gothenburg, Gothenburg, Sweden. [12]Dementia Research Centre, Queen Square Institute of Neurology, University College London, London, UK. ✉e-mail: Nicolai.franzmeier@med.uni-muenchen.de

Aβ plaques induce aberrant activity in surrounding synapses and may therefore promote tau spread along neural pathways[9–11]. Thus, Aβ-related synaptic changes may be associated with accelerated tau spreading along connected brain regions in AD.

The growth-associated protein GAP-43, also known as neuromodulin, is a presynaptic protein strongly involved in synaptic plasticity and neuronal development that is expressed in medial temporal lobe regions that are particularly vulnerable to earliest AD-associated tau pathology[12]. GAP-43 has been found to play a key role in axonal growth and the formation of new synaptic connections[13]; its expression is associated with neuronal activity and is upregulated in rodent models of epilepsy with strong hyperexcitatory neuronal activity[14]. In AD, earlier studies have consistently found increased levels of GAP-43 in cerebrospinal fluid (CSF)[15–17], suggesting that CSF GAP-43 may capture Aβ-related deviations in synaptic integrity and activity. Furthermore, higher CSF levels of GAP-43 have been associated with worse cognitive performance and faster symptom worsening in AD, indicating that GAP-43 may be a proxy of synaptic dysfunction that parallels cognitive decline in the disease[16,18].

Given the role of GAP-43 in synaptic remodeling and activity and the consistent finding of increased levels of CSF GAP-43 in AD, we hypothesize that elevated GAP-43 and associated synaptic changes may be linked to facilitated trans-synaptic spread of tau pathology in AD. However, this hypothesis has hitherto not been systematically tested in AD patients. To investigate the relationship between Aβ-related synaptic changes and tau spread across connected brain regions, we examined cerebrospinal fluid (CSF) levels of growth-associated protein 43 (GAP-43) in a cohort of 93 patients from the Alzheimer's disease neuroimaging initiative (ADNI) dataset, covering the AD spectrum from cognitively normal to dementia as well as Aβ-negative cognitively normal controls. All subjects underwent baseline Aβ-PET and CSF GAP-43 assessments together with longitudinal Flortaucipir tau-PET over the course of approximately three years to robustly determine tau accumulation over time. We further obtained resting-state fMRI connectivity templates from an independent sample of 42 healthy ADNI controls without evidence of Aβ or tau pathology to determine the healthy brain's connectome along which we modeled tau spread[5,19,20]. Using these data, we report that higher baseline CSF GAP-43 levels are associated with accelerated Aβ-related tau accumulation and that higher CSF GAP-43 levels specifically increase the Aβ-related spread of tau pathology from local epicenters across functionally connected brain regions. Addressing these questions is key to better understanding how Aβ and synaptic changes may conjointly drive tau aggregation and spread, which ultimately triggers neurodegeneration and the development of dementia symptoms.

## Results

We included a total of 39 Aβ-negative CN controls, 33 CN Aβ+ and 21 Aβ+ patients with AD clinical syndrome (i.e., 18 MCI, 3 AD dementia) from the Alzheimer's disease neuroimaging initiative (ADNI) database. All subjects underwent baseline 18F-Florbetapir amyloid-PET to determine the global amyloid-PET load in centiloids, CSF assessments of GAP-43, as well as longitudinal 18F-Flortaucipir tau-PET with an average follow-up time of 3.21 ± 1.48 years. Tau-PET images were parcellated into 200 cortical ROIs of the Schaefer atlas[21], and intensity normalized to the inferior cerebellar gray. Longitudinal tau-PET change rates were determined per ROI using linear mixed models with random slope and intercept in line with our previous studies[20,22,23]. Group demographics and clinical characteristics are shown in Table 1. Group demographics of the healthy control sample used to determine the connectome template can be found in Supplementary Table 1. Brain surface renderings of baseline tau-PET SUVR and longitudinal tau-PET change rates are shown in Fig. 1, illustrating elevated cross-sectional temporoparietal tau-PET SUVRs and longitudinal tau-PET increase in 54 Aβ+ patients on the AD spectrum patients vs. absent tau accumulation in the 39 CN Aβ− controls.

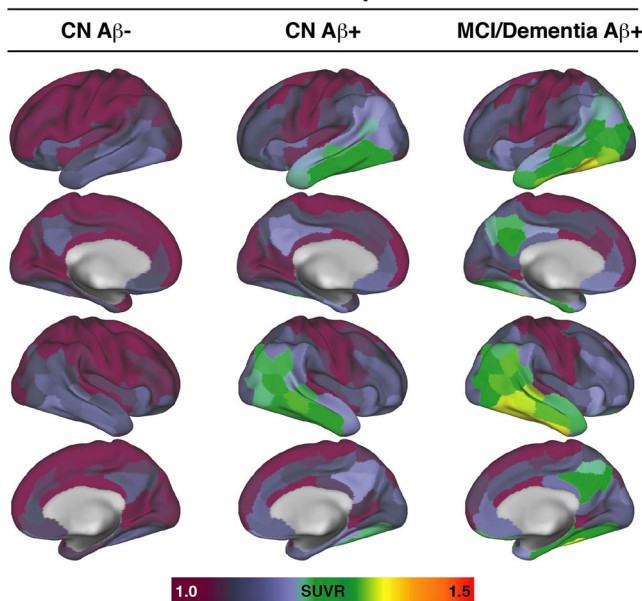

***Baseline Flortaucipir tau-PET***

| CN Aβ- | CN Aβ+ | MCI/Dementia Aβ+ |

1.0 — SUVR — 1.5

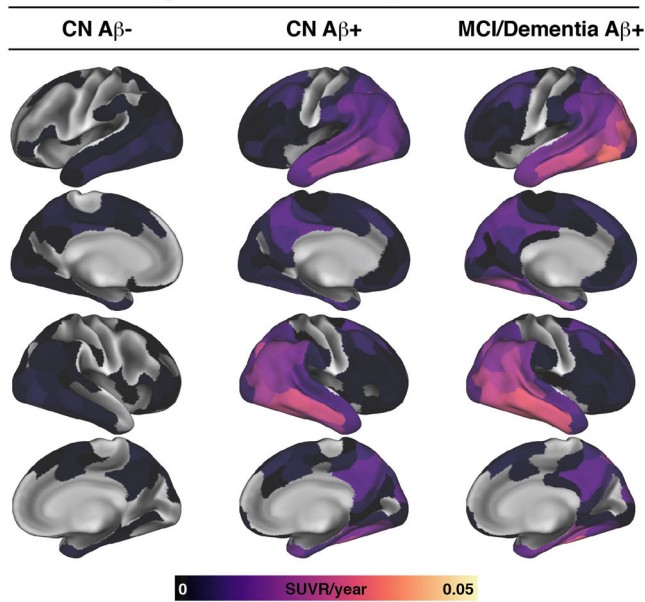

***Longitudinal Flortaucipir tau-PET***

| CN Aβ- | CN Aβ+ | MCI/Dementia Aβ+ |

0 — SUVR/year — 0.05

**Fig. 1 | Tau-PET.** Baseline tau-PET and longitudinal tau-PET change rates stratified by diagnostic groups. CN Cognitively Normal, MCI Mild Cognitive Impairment, PET Positron Emission Tomography, SUVR Standardized uptake value ratio, Aβ amyloid-beta.

## Higher CSF GAP-43 is linked to faster Aβ-related global and temporal lobe tau accumulation

In the first step, we tested whether higher Aβ was associated with faster tau accumulation in pre-defined summary ROIs that are typically used to quantify tau-PET[24] and whether this association was stronger at higher CSF GAP-43 levels. To this end, we used linear regression models and computed the CSF GAP-43 x centiloid interaction on annual tau-PET change rates using global tau-PET and the temporal meta-ROI as pre-defined readouts. As hypothesized, we found consistent and significant CSF GAP-43 x centiloid interactions for global ($p = 0.009$, Fig. 2A) and temporal meta-ROIs ($p = 0.007$, Fig. 2B), where higher CSF GAP-43 was associated with a stronger association between centiloid and tau-PET increase over time, suggesting that higher CSF GAP-43 is indeed linked to faster Aβ-related tau accumulation.

**Table 1 | Sample characteristics**

|  | CN Aβ − (n = 39) | CN Aβ + (n = 33) | MCI/Dementia Aβ+ (n = 21) | p-value |
|---|---|---|---|---|
| Age | 72.8 ± 5.06 | 76.6 ± 6.36 | 77.9 ± 7.06 | 0.003 |
| Sex (female/male) | 29/19 | 23/10 | 8/13 | 0.063 |
| Centiloid | 8.67 ± 7.12 | 67.6 ± 41.8 | 69.2 ± 32.5 | <0.001 |
| CSF GAP-43 | 4780 ± 2220 | 5570 ± 3820 | 5560 ± 3070 | 0.473 |
| CSF p-tau$_{181}$ | 19.1 ± 6.79 | 27.1 ± 13.3 | 31.5 ± 18.6 | <0.001 |
| Tau global SUVR | 1.05 ± 0.070 | 1.11 ± 0.092 | 1.13 ± 0.137 | 0.002 |
| MMSE | 29.1 ± 1.05 | 28.6 ± 1.85 | 26.3 ± 2.89 | <0.001 |
| ADAS13 | 7.52 ± 4.14 | 8.35 ± 5.56 | 16.9 ± 8.94 | <0.001 |
| Tau-PET follow-up years | 3.72 ± 1.58 | 2.93 ± 1.23 | 2.71 ± 1.42 | 0.014 |
| Number of tau-PET visits | 2.79 ± 1.13 | 2.88 ± 0.893 | 2.81 ± 0.928 | 0.491 |

Group comparisons were assessed using ANOVAs for continuous or Chi-squared tests for categorical variables, using two-sided alpha thresholds of 0.05 without adjustments for multiple comparisons. For Continuous measures, means and standard deviations are shown, for categorical measures, absolute numbers are shown.

*MMSE* Mini Mental State Exam, *ADAS13* Alzheimer's disease Assessment scale, *CSF* cerebrospinal fluid, *p-tau* phospho tau 181, *GAP-43* growth-associated protein 43, *PET* positron emission tomography.

Detailed model statistics are shown in Table 2. All models were controlled for age, sex, diagnosis, and CSF p-tau$_{181}$.

## Elevated CSF GAP-43 is associated with faster amyloid-related tau spreading across connected brain regions

Next, we specifically investigated whether elevated CSF GAP-43 was linked to stronger tau accumulation in brain regions closely connected to a given patient's tau epicenter (i.e., regions in which tau pathology is assumed to emerge first), and whether this effect weakened across regions that are less connected to the tau epicenter to recapitulate connectivity-associated tau spread across the brain. To this end, we modeled individual tau epicenters as 5% of brain regions with the highest baseline tau-PET signal and grouped the remaining brain regions into four quartiles (i.e., Q1-4) depending on their average template-based connectivity strength to the subject-specific tau epicenters. Probability mappings of tau epicenters and Q1-Q4 ROIs are shown in Fig. 3A, showing that tau epicenters fall predominantly in the inferior temporal lobe and that Q1 ROIs with the closest connection to the epicenters cover temporoparietal and frontal tau vulnerable regions. In contrast, Q4 ROIs with the least connectivity to the tau epicenter cover primary visual and sensorimotor cortices, i.e., regions that accumulate tau very late in AD. When using these subject-specific Q1-Q4 ROIs to determine the effects of CSF GAP-43 on Aβ-related tau accumulation, we, again, detected a significant centiloid x CSF GAP-43 interaction for Q1 ($p = 0.004$, Fig. 3B) and Q2 ($p = 0.009$, Fig. 3C), which became statistically non-significant for Q3 ($p = 0.080$, Fig. 3D) and Q4 ($p = 0.344$, Fig. 3E). In line with these results, the variance explained by the models (i.e., marginal $R^2$) in annual tau-PET change rates, gradually weakened across Q1-Q4 ROIs (see Table 2). The same analyses were repeated across different connectivity density thresholds for modeling tau spread, yielding consistent results (Supplementary Table 2). As for the models on global and temporal meta ROI tau-PET, all linear models were controlled for age, sex, diagnosis, p-tau$_{181}$. Together, these analyses suggest that CSF GAP-43 is linked to stronger Aβ-related tau spreading from regional tau epicenters to the most strongly connected regions, whereas this effect weakened across regions that were less connected to the tau epicenter, suggesting that

GAP-43 specifically drives the spread of tau across connected brain regions.

To further illustrate the effect of CSF GAP-43 on tau accumulation rates across the Q1-Q4 ROIs, we compared ROI-specific tau-PET change rates stratified by high vs. low CSF GAP-43 defined by median split in the subsample of 54 Aβ+ subjects that are on the AD trajectory (Fig. 4A). In line with the previous analyses, we found consistently higher tau-PET ROCs in the Aβ+ subjects with above median CSF GAP-43 levels vs. Aβ+ subjects with below median CSF GAP-43 levels for Q1 (F = 10.592, $p = 0.0022$, Cohen's d = 0.805), Q2 (F = 10.135, $p = 0.0027$, Cohen's d = 0.793), Q3 (F = 10.004, $p = 0.0028$, Cohen's d = 0.841) and Q4 (F = 6.217, $p = 0.0165$, Cohen's d = 0.619).

## Higher CSF GAP-43 is associated with a stronger association between tau epicenter connectivity and tau accumulation patterns

Finally, we investigated whether higher CSF GAP-43 was specifically related to faster connectivity-associated tau spreading in the presence of elevated Aβ. To this end, we assessed the template-based functional connectivity pattern of subject-specific tau epicenters (i.e., 5% of ROIs with the highest baseline tau-PET) to the remaining 95% of brain regions and tested whether the epicenter connectivity pattern was associated with the tau-PET accumulation pattern in the rest of the brain using linear regression for each subject. We hypothesized that brain regions that are more closely connected to the tau epicenters (i.e., which show shorter connectivity-based distance) exhibit greater tau-PET change than regions that are more weakly connected to the tau epicenters (i.e., which show higher connectivity-based distance), resulting in a negative regression slope. As expected, the resulting regression-derived beta-values were overall negative (T = −7.8103, $p < 0.001$, indicating that regions closely connected to subject-specific tau epicenters have faster tau accumulation) and became more negative at higher centiloid levels, suggesting that connectivity-associated tau spread can be observed particularly at higher Aβ levels (B = −0.00351, CI = [−0.00537;−0.00165], $p < 0.001$). However, we found a significant centiloid x CSF GAP-43 interaction on these beta-values of connectivity-associated tau spread, where higher CSF GAP-43 strengthened the association between Aβ and epicenter connectivity-associated tau accumulation (Fig. 4B, $p = 0.037$). These findings suggest that higher CSF GAP-43 is associated with stronger Aβ- and connectivity-related tau spread. As done previously, all models were controlled for age, sex, diagnosis and p-tau$_{181}$.

## Discussion
The key aim of this combined neuroimaging and biofluid biomarker study was to investigate whether synaptic changes are associated with accelerated tau spreading in AD, thereby contributing to disease progression. Supporting this, our main findings were that higher levels of the presynaptic protein GAP-43 in the CSF typically increased in AD patients[18,25], are associated with faster Aβ-related tau accumulation and spreading across connected brain regions. Specifically, the effect of Aβ on tau accumulation was increased at higher CSF GAP-43 levels, particularly in those brain regions that are most closely connected to patient-specific epicenters of highest baseline tau pathology. Further, the connectivity patterns of tau epicenters aligned more closely with longitudinal tau accumulation patterns in patients with both high PET-assessed Aβ and CSF GAP-43, suggesting that elevated CSF GAP-43 specifically promotes the Aβ-associated spreading of tau from local epicenters across connected regions. Together, our findings indicate that higher CSF GAP-43 is associated with faster Aβ-related tau accumulation and connectivity-mediated spread in AD, rendering synaptic changes as a potential target for attenuating tau accumulation in AD.

Our key finding of higher GAP-43 being related to accelerated tau accumulation in AD supports a mechanistic model of AD pathophysiology in which synaptic changes are critically involved in the

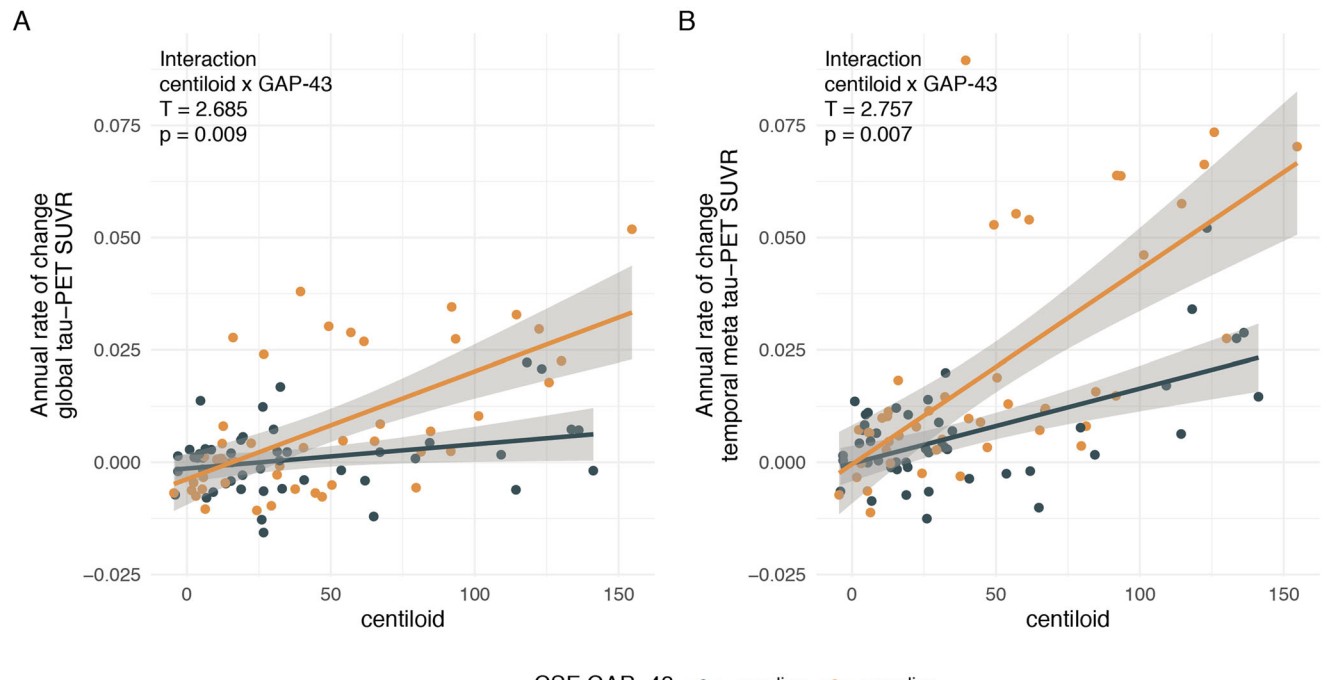

CSF GAP–43 <median >median

**Fig. 2 | GAP-43 is associated with faster amyloid-related tau accumulation in meta ROIs.** Scatterplots illustrating the interaction between amyloid-PET (i.e., centiloid) and CSF GAP-43 levels on tau-PET changes in a global cortical ROI (**A**), as well as in a temporal meta ROI (**B**). Regression models were corrected for age, sex, diagnosis, and CSF p-tau$_{181}$. Note that all interactions were computed using continuous GAP-43 measures across the entire study cohort ($N = 93$), median split was only performed for visualization. β-values reflect standardized regression weights. All T- and two-sided *p*-values were derived from linear regression. Linear model fits (i.e., least squares line) are indicated together with 95% confidence intervals displayed as error bands. Source data are provided as a Source Data file. PET Positron Emission Tomography, SUVR Standardized uptake value ratio, CSF cerebrospinal fluid, GAP-43 Growth-associated protein 43.

**Table 2 | Centiloid x CSF GAP-43 interaction**

| | Tau-PET ROC ROI | Estimate | 95%CI | T-value | *P* | Model marginal R² |
|---|---|---|---|---|---|---|
| Pre-defined ROIs | Global | 0.0002 | 0.0000–0.0003 | 2.685 | 0.009 | 0.378 |
| | Temporal meta | 0.0003 | 0.0001–0.0005 | 2.757 | 0.007 | 0.524 |
| Personalized ROIs | Q1 | 0.0002 | 0.0001–0.0004 | 2.926 | 0.004 | 0.404 |
| | Q2 | 0.002 | 0.000–0.003 | 2.661 | 0.009 | 0.363 |
| | Q3 | 0.0001 | −0.00001 to 0.0002 | 1.771 | 0.080 | 0.282 |
| | Q4 | 0.0000 | −0.00001 to 0.0001 | 0.952 | 0.344 | 0.231 |

Statistical indices were derived from linear regression models and displayed the centiloid by CSF GAP-43 interaction on tau-PET change rates in different ROIs.
*GAP-43* growth-associated protein 43, *PET* positron emission tomography, *CI* confidence interval.

Aβ-associated spreading of tau pathology, i.e., the key driver of neurodegeneration and cognitive decline in AD[2,26,27]. This model builds on in vitro, animal and post-mortem data, showing that (1) tau spreads across synapses in an activity-dependent manner[9,10,28–30] and (2) Aβ induces synaptic hyperactivity and synaptic dysfunction by attenuating glutamate re-uptake and reducing the sensitivity to GABA[31,32]. Congruently, patient studies have reported a higher prevalence of subclinical epileptiform brain activity related to Aβ deposition and *APOE* ε4 carriage[33,34] as well as Aβ-related hyperactivity and hyperconnectivity on EEG and resting-state fMRI[14,35], providing converging evidence that Aβ induces a hyperexcitatory shift in neuronal activity. GAP-43 has been implicated in presynaptic vesicle cycling and its expression and serum levels are upregulated in hyperexcitatory conditions such as epilepsy[14,35] and GAP-43 CSF levels increase in AD already at preclinical stages[18,25]. Further, inhibition of GAP-43 has been shown to drastically reduce synaptic glutamate release[36,37], together supporting a key role of GAP-43 in neurotransmitter release and synaptic activity, which might be exacerbated in AD[38]. Thus, CSF GAP-43 increases in AD may mirror Aβ-induced hyperexcitatory synaptic changes[39], yet this remains to be specifically tested in future studies combining electrophysiological measures of neuronal activity with soluble GAP-43 measures in AD patients and/or AD model systems.

Aβ-related synaptic changes toward hyperexcitatory activity may putatively drive tau spreading in AD since higher neuronal activity has been shown to induce faster neuronal tau secretion, ensuing transsynaptic propagation of seeding competent tau in vitro and in animal models[3,40–42]. Similarly, AD patient studies have shown that hyperphosphorylated tau (p-tau) is actively secreted to the CSF in the presence of Aβ[43], potentially due to Aβ-induced neuronal hyperexcitability[10,11]. Further, we found previously that higher CSF p-tau predicts faster spreading of tau across connected brain regions, supporting the view that secretion of Aβ-related p-tau contributes to tau spreading[20,44]. Our current results extend these previous findings and provide further support for a specific role of synaptic changes in connectivity-dependent tau spreading using actual AD patient imaging and biomarker data. In particular, our findings of elevated GAP-43 being specifically related to Aβ-associated tau accumulation in brain regions that were most strongly connected to the initial temporal lobe

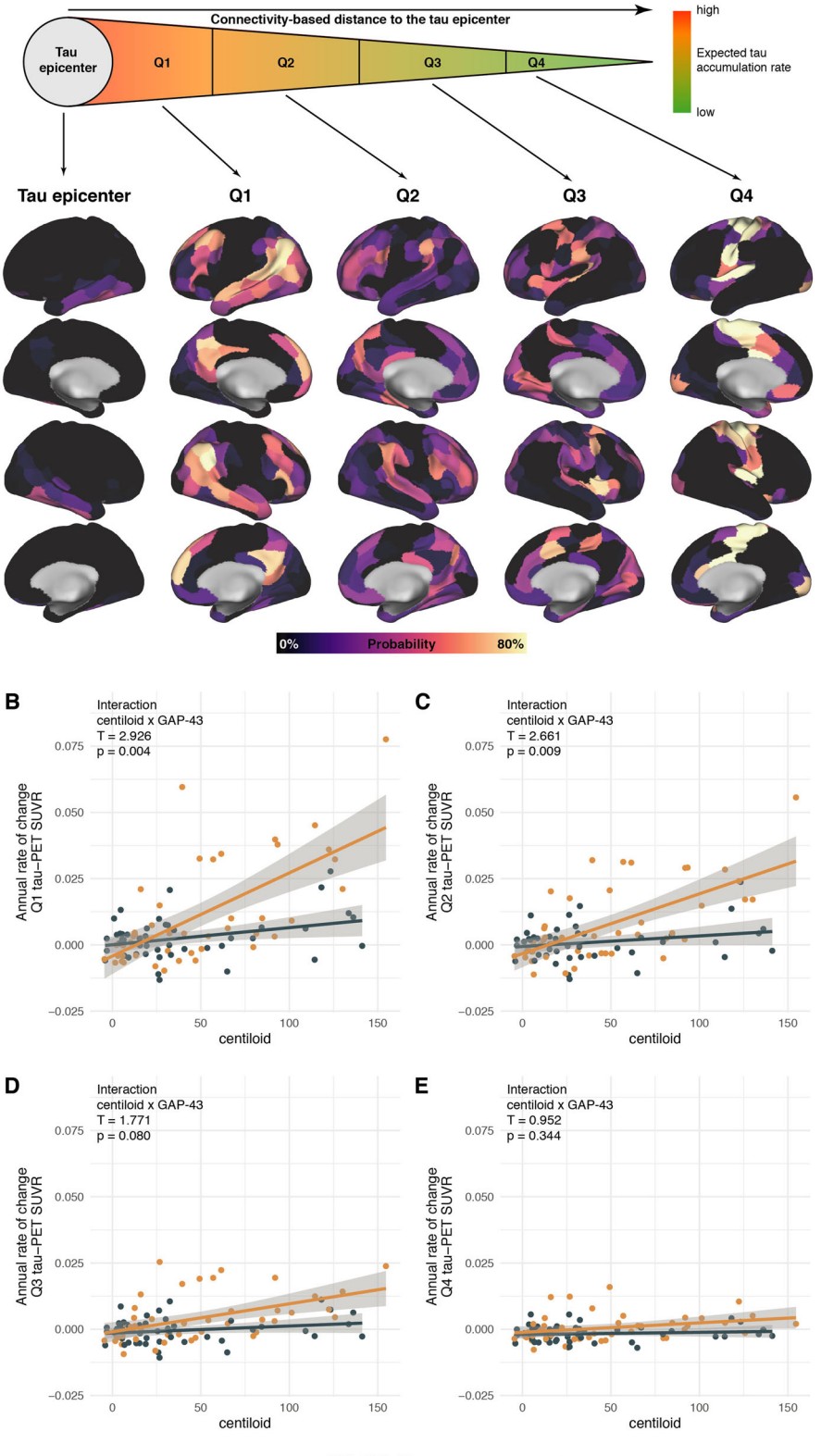

**A** *Subject-specific modeling of longitudinal tau accumulation*

epicenters of tau pathology suggest that synaptic changes do not simply lead to brain-wide and diffuse tau accumulation but specifically facilitate tau spreading across connected brain regions[18]. Of note, all associations between CSF GAP-43 and faster Aβ-related tau accumulation and spread were statistically independent of p-tau$_{181}$, which we previously showed to mediate the association between Aβ and tau

accumulation[20]. CSF p-tau levels are correlated with CSF GAP-43[18], which further supports the view that GAP-43-related synaptic changes may foster the secretion of p-tau in AD. Here, an important next step will be to further determine the complex timing and interplay of p-tau and synaptic biomarker changes, PET-assessed tau accumulation and changes in neuronal activity and connectivity[45]. As an alternative

**Fig. 3 | GAP-43 is associated with faster amyloid-related tau spreading.** Illustration of determining subject-specific ROIs for tau accumulation using tau epicenter-based connectivity (**A**). Scatterplots illustrating the interaction between Aβ-PET (i.e., centiloid) and CSF GAP-43 on tau-PET changes in connectivity-derived ROIs across the entire study cohort (*N* = 93), ranging from regions that are closely connected to subject-specific tau epicenters (Q1, Panel **B**) to regions that are less strongly connected to subject-specific tau epicenters (Q2-Q4, Panels **C**–**E**). Tau epicenters were defined as 5% of brain regions with the highest baseline tau-PET SUVRs. Connectivity was derived using a connectome template based on 3T multi-

band resting-state fMRI data from healthy controls. Regression models were corrected for age, sex, diagnosis, and CSF p-tau$_{181}$. Note that all interactions were computed using continuous GAP-43 measures, median split was only performed for visualization. β-values reflect standardized regression weights. All T- and two-sided *p*-values were derived from linear regression. Linear model fits (i.e., least squares line) are indicated together with 95% confidence intervals as error bars. Source data are provided as a Source Data file. PET Positron Emission Tomography, SUVR Standardized uptake value ratio, CSF cerebrospinal fluid, GAP-43 Growth-associated protein 43.

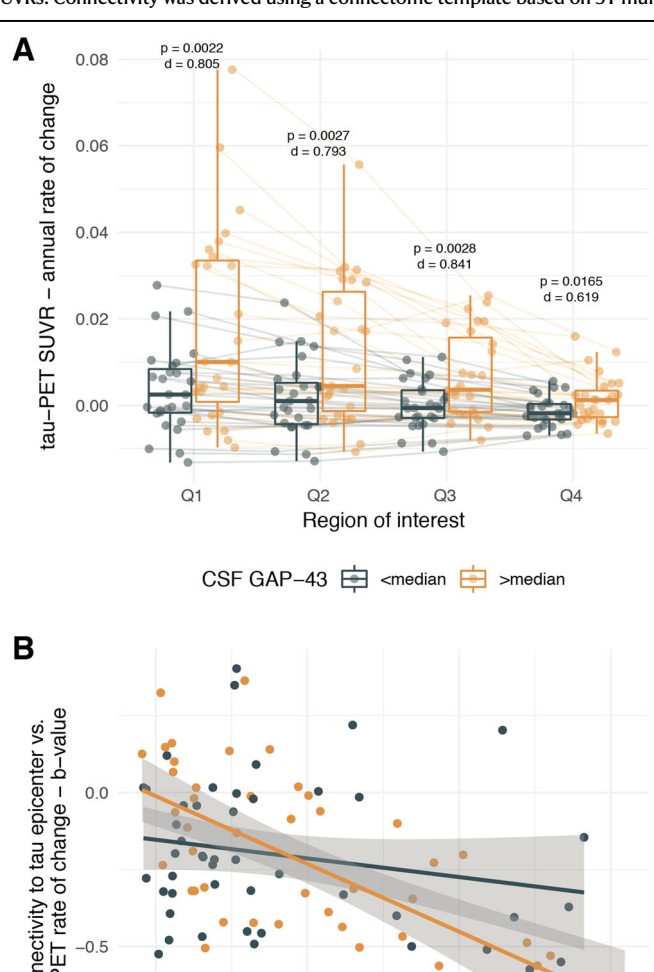

explanation, GAP-43 and p-tau may mirror higher Aβ-related neuronal activity and metabolism, which may lead to overall higher transcriptional and translational activity of GAP-43 and tau, ensuing faster local tau aggregation independent of trans-synaptic spread[46]. Yet, our findings of GAP-43, specifically promoting the spread of tau across connected brain regions, speak against this hypothesis. Alternatively, GAP-43 may also passively increase following AD-related synaptic degeneration and therefore parallel tau spreading. However, synaptic biomarker studies have shown simultaneous increases and decreases of different synaptic proteins, suggesting that synaptic biomarker changes are more complex[47]. Thus, it will be important for future studies to address these open questions and potential alternative explanations in order to develop a better mechanistic understanding of AD pathophysiology and progression beyond Aβ and tau and to identify potential therapeutic targets to prevent tau spreading.

Several limitations should be considered when interpreting the results of the current study. First, our findings are limited to the synergistic contribution of synaptic marker CSF GAP-43 abnormalities on Aβ-related tau accumulation, while CSF increases in numerous other synaptic markers have been reported previously in AD, including SNAP-25, synaptotagmin-1 or neurogranin[25,47]. We specifically focused on GAP-43 due to its strong presynaptic expression, which might be particularly relevant for the putative anterograde direction of tau spreading[3] and due to its role in synaptic signaling and neuronal activity[14,35], which we considered critical for translating in vitro findings of activity-dependent tau spreading across connected neurons to human data[3,40–42,48]. Nevertheless, a breadth of other CSF synaptic biomarker increases have been shown in AD patients but their specific mechanistic meaning and contribution to AD pathophysiology remains elusive, including a dedicated analysis of the underlying mechanisms that drive CSF GAP-43 increases in AD[15,16,18,47,49]. Also, no other synaptic biomarker beyond GAP-43 is currently available in close enough proximity to longitudinal tau-PET assessments in open-access datasets such as ADNI, which clearly limits the study of synaptic changes in tau spreading in AD. Measuring these biomarkers in future studies or in potentially available ADNI biosamples would be critical since we believe that synaptic CSF or PET imaging biomarkers hold high potential to specifically determine the role and contribution of presynaptic and post-synaptic changes to the progression of AD pathophysiology, neurodegeneration and symptom manifestation. Specifically, the combination of multiple markers of synaptic changes and primary AD pathophysiology (i.e., Aβ and tau) will be critical in determining a more mechanistic disease model and help identify whether and how amyloid-associated pre- or post-synaptic changes contribute to tau spreading and how tau induces synaptic degeneration[47]. To this end, it will be particularly important to also elucidate the exact molecular and biological meaning of synaptic biomarkers and their respective increases/decreases in biofluids or on PET imaging to allow drawing biological conclusions about their specific involvement in the AD pathophysiological cascade[50]. Of note, our study did not report significant increases in CSF GAP-43 related to Aβ positivity in the current study despite numerically higher GAP-43 levels in Aβ+ subjects in the current sample. This lack of statistical significance is potentially related to the lower sample size (*N* = 93) of the current study compared to a previous study reporting this group

**Fig. 4 | Tau spreading across connected regions is accelerated at higher GAP-43 levels.** Boxplots illustrating the effect of GAP-43 levels on tau-PET increase in Q1-Q4 ROIs in the Aβ+ subjects (*N* = 54) (**A**). Scatterplot (**B**), illustrating the interaction between amyloid-PET (i.e., centiloid) and CSF GAP-43 on connectivity-based tau spreading (*y*-axis) in the entire study cohort (*N* = 93). Regression and ANCOVA models were corrected for age, sex, diagnosis and CSF p-tau$_{181}$. Note that interactions for Panel **B** were computed using continuous GAP-43 measures and that the median split was performed only for visualization. β-values reflect standardized regression weights. All T- and two-sided *p*-values were derived from linear regression. Linear model fits (i.e., least squares line) are indicated together with 95% confidence intervals. Boxplots are displayed as median (center line) ± interquartile range (box boundaries) with whiskers including observations falling within the 1.5 interquartile range. Source data are provided as a Source Data file. PET Positron Emission Tomography, SUVR Standardized uptake value ratio, CSF cerebrospinal fluid, GAP-43 Growth-associated protein 43.

difference in the ADNI cohort ($N = 786$)[18]. In addition, it will be critical to validate our findings in other datasets, yet no suitable replication dataset was available to the authors at the time of data analysis. Further, we used a resting-state fMRI template based on healthy controls to model tau spreading in line with our previous studies[20,51], since high-quality resting-state fMRI data was not consistently available across all patients included in the current study. Therefore, we were not able to determine whether CSF GAP-43 increases may explain previous reports of Aβ-related hyperconnectivity and whether increases in inter-regional connectivity promote the spreading of tau[52]. However, once larger AD patient datasets with high-quality resting-state fMRI and biomarker data become available, addressing this question will be critical in helping us better understand how Aβ, synaptic and connectivity changes relate to tau spreading.

In conclusion, the current results suggest a key involvement of synaptic changes related to GAP-43 in Aβ-related tau spreading. Our results highlight that the link between Aβ and tau spreading may be linked to synaptic changes and potentially synaptic activity, rendering synaptic changes as promising targets for further research[45,50]. Ultimately, synapses are considered the pathway for the spreading of tau pathology in AD[3], hence a better understanding of how synapses and synaptic changes contribute to this process will be critical to develop targeted therapies for attenuating tau spreading to prevent downstream neurodegeneration and cognitive decline.

## Methods

All research complies with ethical regulations for human subjects. Ethical approval was obtained by each respective ADNI site, and written informed consent was collected from all participants in accordance with the declaration of Helsinki.

### Sample

We included 93 ADNI participants based on the availability of longitudinal flortaucipir tau-PET (i.e., at least one follow-up flortaucipir-PET visit), baseline florbetapir Aβ-PET and baseline CSF GAP-43 measures. All subjects were classified as Aβ-positive or -negative (Aβ+/−) based on an established global florbetapir Aβ-PET threshold (SUVR > 1.11)[53]. ADNI investigators assessed and diagnosed subjects as either cognitively normal (CN; Mini Mental State Examination [MMSE] ≥ 24, Clinical Dementia Rating [CDR] = 0, non-depressed), mildly cognitively impaired (MCI; MMSE ≥ 24, CDR = 0.5, objective memory-impairment on education-adjusted Wechsler Memory Scale II, preserved activities of daily living) or demented (MMSE = 20-26, CDR > 0.5, NINCDS/ADRDA criteria for probable AD). The sample included 39 Aβ− CN subjects and 54 Aβ+ individuals covering the AD spectrum: (CN/MCI/Dementia $n = 33/18/3$), Aβ− subjects with a diagnosis other than CN were excluded owing to suspected non-AD pathology (i.e., SNAP).

### CSF measurements

CSF GAP-43 concentration was measured at the University of Gothenburg using an in-house enzyme-linked immunosorbent assay as previously described in detail[15,18]. CSF p-tau$_{181}$ levels were determined using the Elecsys system, as previously reported for ADNI[54].

### MRI and PET acquisition

Structural MRI was acquired using 3T Siemens, GE and Philips scanners. T1-weighted structural scans were collected using an MPRAGE sequence (TR = 2300 ms; Voxel size = 1 × 1 × 1 mm); 10-min Resting-state fMRI data was acquired on 3T Siemens scanners (TR/TE = 3000 ms/90 ms). PET data was assessed post intravenous injection of $^{18}$F-labeled tracers flortaucipir (collection of 6 × 5 min time-frames, 75–105 min post-injection) and florbetapir (collection of 4 × 5 min time-frames, 50–70 min post-injection); for detailed information, see http://adni.loni.usc.edu/methods/pet-analysis-method/pet-analysis/.

### Neuroimage processing

All images were screened for artifacts before preprocessing. T1-weighted structural MRI scans were bias-corrected, segmented into gray matter, white matter and cerebrospinal fluid segments, and non-linearly warped to Montreal Neurological Institute (MNI) space using the CAT12 toolbox (https://neuro-jena.github.io/cat12-help/). Dynamically acquired PET images were realigned and averaged to obtain single flortaucipir/florbetapir images, which were rigidly registered to the T1-weighted MRI scan. Reference regions (i.e., inferior cerebellar gray for flortaucipir, whole cerebellum for florbetapir)[55], the cortical Schaefer atlas including 200 regions of interest (ROIs) were warped from MNI to T1-native space using the CAT12-derived non-linear normalization parameters, masked with subject-specific gray matter and applied to PET data to determine standardized uptake value ratios (SUVRs) for each region of the Schaefer 200 atlas[21]. Tau-PET in the pre-defined temporal meta ROI was determined as the mean of Braak stages I, III and IV, based on our previously established mapping of Schaefer ROIs to Braak-stage ROIs (see Supplementary Data 1)[5,56]. Global florbetapir SUVRs were converted to centiloid using equations provided by ADNI. To determine longitudinal tau-PET change for each ROI, we employed linear mixed models with tau-PET SUVRs as the dependent variable and time from baseline as the independent fixed effect controlling for random slope and intercept as described previously[20,22,23].

### Assessment of a functional connectivity template

For the independent sample of 42 cognitively normal controls (see Supplementary Table 1 for demographics) that were used to generate the connectome template, resting-state fMRI (i.e., EPI) images were slice-time and motion corrected (i.e., realigned to the first volume) and co-registered to their respective T1-weighted images. Using rigid-transformation parameters, T1-derived gray matter, eroded white matter and eroded cerebrospinal fluid (CSF) segments were transformed to EPI space. To denoise EPI images, we regressed out nuisance covariates (i.e., eroded white matter and eroded CSF time series plus six motion parameters) and applied detrending and band-pass filtering (0.01–0.08 Hz) in EPI native space. To further reduce movement artifacts that may compromise connectivity assessment[57], we performed motion scrubbing in which volumes exceeding a 0.5 mm frame-wise displacement threshold were removed, as well as one prior and two subsequent volumes. All subjects had at least 5 min of resting-state fMRI remaining after scrubbing[58]. Spatial smoothing was not carried out to avoid artificially enhancing functional connectivity caused by signal spilling between adjacent brain regions. Pre-processed rs-fMRI images were subsequently warped to MNI space using the CAT12-derived spatial normalization parameters.

Subject-specific functional connectivity matrices were determined across the 200 ROIs of the Schaefer atlas as Fisher-z-transformed Pearson moment correlations between ROI-specific time series. All individual matrices were averaged and thresholded at 30% density, with negative connections excluded, following our previously established protocol to maximize consistency across studies[5,23]. The average functional connectivity was then converted to a distance-based connectivity matrix[59], where shorter path-lengths between ROIs represent stronger connectivity, in line with our previous work[5,20]. Note that analyses were repeated across different density thresholds between 10 and 40%, which yielded fully consistent results (Supplementary Table 2).

### Assessment of connectivity-mediated tau spreading

To determine connectivity-associated tau spread, we employed a previously established approach, defining subject-specific tau epicenters as the 10 ROIs (i.e., 5% of ROIs) with the highest baseline tau-PET SUVRs[20]. For each subject, we then assessed the seed-based connectivity of these 10 ROIs to the remaining 190 ROIs using the

connectome template and determined the regression-based association between connectivity of the subject-specific tau epicenter and tau-PET rate of change in the 190 non-epicenter regions as a metric of connectivity-associated tau spread[20]. Negative β-values were expected, meaning that stronger connectivity (represented as smaller values given that connectivity measures were converted to distance-based) would be associated with greater tau-PET change. From this connectivity-based analysis, for each participant, we further grouped the 190 non-epicenter regions into quartiles based on their connectivity to the tau epicenters (Q1 to Q4). The top 25% of regions with the highest connectivity to the tau epicenters were part of Q1, the following 25% in Q2, etc. The average tau-PET rate of change in each quartile was calculated and used in further statistical analyses[8,20,51].

### Statistics

All analyses were performed using R statistical software (Version 4.0.4). Group demographics were compared between diagnostic groups (i.e., CN Aβ−, CN Aβ+, AD clinical syndrome) using ANOVAs for continuous measures and Chi-squared tests for categorical data. Subject-specific tau-PET change rates were determined by fitting linear mixed models to ROI-specific tau-PET data, with time from baseline as a predictor adjusting for random slope and intercept[20,22,23]. To test whether higher CSF GAP-43 is associated with accelerated Aβ-related tau accumulation, we used linear regression models to compute the CSF GAP-43 x centiloid interaction on annual tau-PET change rates for the previously described temporal meta as well as global cortical ROIs, controlling for age, sex, diagnosis and CSF p-tau$_{181}$ (i.e., R linear model equation: Tau-PET change ~ CSF GAP-43 x centiloid + age + sex + diagnosis + CSF p-tau$_{181}$)[56].

Next, we examined whether elevated CSF GAP-43 was linked specifically to connectivity-associated tau accumulation and spreading. To this end, we assessed the CSF GAP-43 x centiloid interaction on tau-PET change rates in the personalized Q1-Q4 ROIs (i.e., same equation as above using tau-PET change in Q1 to Q4 ROIs as the dependent variable). In the subset of Aβ+ participants, we further compared tau-PET change rates within Q1-Q4 ROIs between subjects with above and below median CSF GAP-43 using ANCOVAs and assessed Cohen's d-based effect sizes between above vs. below median CSF GAP-43 groups (i.e., R ANCOVA equation: Tau-PET change ~ CSF GAP-43 group [defined by median split] + centiloid + age + sex + diagnosis + CSF p-tau$_{181}$).

Lastly, we tested whether tau epicenter connectivity was more predictive of tau accumulation patterns at higher centiloid and CSF GAP-43 levels. To this end, we first assessed the subject-level association between tau epicenter connectivity and tau accumulation in non-epicenter ROIs (i.e., R linear model equation per subject: Tau-PET change ~ connectivity to tau epicenter) and extracted the beta value of connectivity to determine the degree of connectivity-mediated tau spreading. Next, we determined the effect of centiloid alone (i.e., independent variable) on the regression-derived beta-values (i.e., dependent variable) that reflected the association between tau epicenter connectivity and subject-level tau-PET change rate patterns defined in the previous step, as well as the centiloid x CSF GAP-43 interaction (i.e., independent variable) on the regression-derived beta-values (i.e., R linear model equation 1: beta value ~ centiloid + age + sex + diagnosis + CSF p-tau$_{181}$; equation 2: beta value ~centiloid x CSF GAP-43 + age + sex + diagnosis + CSF p-tau$_{181}$).

All above-described linear mixed models and ANCOVAs were controlled for age, sex, diagnosis, and CSF p-tau$_{181}$, as well as random slope and intercept. We specifically added CSF p-tau$_{181}$ as a covariate in order to adjust for our previously reported findings of higher CSF p-tau being associated with accelerated Aβ-related tau spread[20]. Note that all results were consistent independent of including p-tau$_{181}$ as a covariate, hence we report only results with p-tau$_{181}$ included in the models here. P-values were considered significant at an alpha of 0.05.

### Reporting summary

Further information on research design is available in the Nature Portfolio Reporting Summary linked to this article.

## Data availability

All data used in this manuscript are publicly available from the ADNI database (adni.loni.usc.edu) upon registration and compliance with the data use agreement. Source data are available online https://doi.org/10.6084/m9.figshare.23905410[60].

## Code availability

Example R code with simulated data can be found on GitHub (https://github.com/nfranzme/Published/tree/3352111bbe2b5a157561159909a2ed92bc005994/TauGAP43_CodeRepository).

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

## Acknowledgements

We would like to acknowledge the entire ADNI leadership board and team for providing the data for this study. For a full list of ADNI

investigators, please see https://adni.loni.usc.edu/wp-content/uploads/how_to_apply/ADNI_Acknowledgement_List.pdf. This work was funded by the Hertie Network of Excellence in Neuroscience (awarded to N.F.). H.Z. is a Wallenberg Scholar supported by grants from the Swedish Research Council (#2022-01018 and #2019-02397), the European Union's Horizon Europe research and innovation programme under grant agreement No 101053962, Swedish State Support for Clinical Research (#ALFGBG-71320), the Alzheimer Drug Discovery Foundation (ADDF), USA (#201809-2016862), the AD Strategic Fund and the Alzheimer's Association (#ADSF-21-831376-C, #ADSF-21-831381-C, and #ADSF-21-831377-C), the Bluefield Project, the Olav Thon Foundation, the Erling-Persson Family Foundation, Stiftelsen för Gamla Tjänarinnor, Hjärnfonden, Sweden (#FO2022-0270), the European Union's Horizon 2020 research and innovation programme under the Marie Skłodowska-Curie grant agreement No 860197 (MIRIADE), the European Union Joint Programme–Neurodegenerative Disease Research (JPND2021-00694), the National Institute for Health and Care Research University College London Hospitals Biomedical Research Centre, and the UK Dementia Research Institute at UCL (UKDRI-1003).

## Author contributions

N.F., data analysis, study concept and design, drafting the manuscript; A.D., data preprocessing, critical revision of the manuscript; A.S., D.B., A.D., S.R., F.W., M.G., data preprocessing, critical revision of the manuscript; M.B., A.M., P.A., B.K., H.Z., M.E., study concept and design, critical revision of the manuscript; M.S., study concept and design, drafting the manuscript.

## Funding

## Competing interests

The authors declare no competing interests.
