## [Peer Review File · Nature Communications]

Elevated CSF GAP-43 is associated with accelerated tau accumulation and spread in Alzheimer's diseaseREVIEWER COMMENTS

Reviewer #1 (Remarks to the Author):

In this study, Franzmeier et al. investigate the influence of CSF GAP-43 levels on the interaction between amyloid beta pathology and tau PET accumulation rates in a sample of individuals with and without amyloid pathology from ADNI. The authors find that elevated GAP43 levels strengthen the association between amyloid beta and tau accumulation rates in functionally connected regions that were reconstructed in an independent control sample. Overall, the study is well carried out, using appropriate methodology. However, additional clarifications in text on the methods used, as well as on the authors' choice of approach are needed.

- The model used to assess whether the association between functional connectivity and tau pattern rates depended on centiloid and/or GAP43 as described in ll. 376 is not clear. Please provide model specifications (e.g. formulas) more clearly.

- In ll.317 the authors refer to previous publications for the model specification to determine tau accumulation rates. However, the linear mixed model was not specified in referred to publications, or may have been missed by this reviewer. Please make sure to include specifications of all (linear mixed) models employed in the study, including covariates, random effects, covariance structure etc., in the manuscript.

- Related to above, the authors write in ll.148, ll.182 and ll.366 that they used linear mixed models for the interaction effect of GAP43 and centiloids on annual tau accumulation rates. I suspect that linear regressions were instead used. Please correct in text. If not, what random effects (intercept/slope) were used here, as there are no repeated/clustered measurements?

- Why was a 30% connectivity density chosen? How were negative correlations handled? The authors should make the reasoning behind their choices clear in the manuscript.

- The authors created a "connectivity template" in an independent sample of cognitively normal controls. Please provide a sample description.

- For longitudinal tau scans, did the authors include only two or more images per individual? This should be made clear in the manuscript.

- In the description for Figure 2 betas are noted, but t values are displayed. Please correct.

Reviewer #2 (Remarks to the Author):

Franzmeier et al present their investigation on the effect of CSF GAP-43 levels in Abeta –positive individuals on the speed and pattern of Tau spread, as measured with PET. The manuscript is well written with clear hypotheses and nicely presented data. I see the main finding as that higher CSF GAP-43 levels are predictive for faster disease progression speed in Abeta-positive individuals after correcting for other factors as age and CSF ptau levels. This is in line with the hypotheses and previous work on GAP-43, a

synaptic biomarker. The authors highlight the relevance of synaptic alterations for AD disease progression. Work on CSF GAP43 and on studying mechanisms of disease progression is timely and of potential significance to better understand and prognosticate AD. References are appropriate. While I appreciate that the paper is so focused, the fact that no other protein markers of (pre)synapse are included can also be a weakness. If markers reflecting different processes would have a differential effect on Tau progression, the case for the synaptic changes as driver for Tau-PET progression would strengthen. Now we cannot exclude that higher GAP-43 levels reflects general higher rates of neuronal loss/degeneration including their synapses. Besides this, I have very limited feedback, because the text, figures etc are very clear. I also think the conclusions are phrased with enough caution. I suggest to include a new paragraph on more alternative hypotheses that could explain the results and what to expect (in future studies) including more markers as a broader discussion.

Minor:

1. Do the authors understand why in their sample the GAP43 levels are not significantly increased in the AD patients (as referred in introduction).
2. CSF Ptau181 (Elecsys?) is not mentioned in the methods

Reviewer #3 (Remarks to the Author):

This manuscript by Franzmeier et al., identified relationships between CSF GAP-43 data and amyloid-beta related tau spread measured by longitudinal PET scans. This is an interesting paper using reasonable approaches to address timely questions in the Alzheimer's disease (AD) field, which largely indicates experience and competence among the authors. Overall, this study was well-designed, and the manuscript is well written. I have a few concerns on the methods and results interpretation, which may be addressed in a revision.

Major comments:

1.The authors claimed that resting-state fMRI data from an independent sample of 42 cognitively normal (CN) controls were used to generate a connectome template. It was not clear to me which independent sample was studied. Did they use the 39 amyloid negative CN subjects from the ADNI? As the number of subjects (42 vs. 39) do not match, I speculate that they were using the HCP or HCP aging (age matched with the ADNI?) data. If so, please clarify why they didn't use the fMRI data from the ADNI, but from another database (as it was not used for replication purpose). How many out of the 93 ADNI subjects have fMRI data? To simulate personalized tau spread, isn't it optimal to use fMRI data from the same subjects of tau PET data? Please also provide an explanation for why data from CN individuals were preferred over patient data in the simulations.

2.On page 15, the authors said that "All individual matrices were averaged and thresholded at 30% density". Was group averaging performed before thresholding? Were there any links with negative correlation values survived after thresholding? If so, how would we estimate/interpret path length with

'negative' correlations (please see PMID: 19819337)? In other words, how would we interpret tau spread via 'negative' links? In addition, were the results sensitive to the choice of link density? Perhaps this can only be answered by validating the results use a range of link density.

3.The authors used the 200-ROI Schaefer atlas to parcellate the PET and fMRI data. However, in line 116, they said that "In a first step, we tested whether higher A β was associated with faster tau accumulation in predefined summary ROIs that are typically used to quantify tau-PET²⁴". I understand that the authors aimed to use functional connectivity to simulate tau spread, so a functional connectivity based parcellation was used for both the PET and fMRI data. However, the study of reference #24 used FreeSurfer's Desikan-Killiany (D-K) atlas (68 ROIs) to parcellate the PET data. So, I was wondering how the authors used 200-ROI Schaefer atlas to match with the 68 D-K ROIs and created the predefined summary ROIs. Please list the names and coordinates of predefined summary ROIs in a table for replication purpose.

4.For some analyses (lines 114 to 126; Fig. 2), it was not clear if both CN and patients' data, or only patients' data were used. Do the colors in Fig. 2 represent groups? Please clarify it.

5.This question is related to the last one. Did amyloid-beta negative CN subjects also show similar relationship between CSF GAP-43 data and tau spread as amyloid-beta positive CN and AD patients? Was this relationship stronger in amyloid-beta positive CN than that in amyloid-beta negative CN? If not, shall we think the identified relationships is specific to AD or shared with other tauopathies (e.g., PART or FTD)?

Minor comments:

1.There are some typos in the manuscript. For example, in the Abstract, "and that synapses could be 'key key' targets for preventing tau spreading in AD". One page 5, "To this end, we 'used linear regression models' and computed the CSF GAP-43 x centiloid interaction on annual tau-PET change rates using global tau-PET and the temporal meta-ROI as pre-defined readouts 'using linear regression models'." Please correct them.

2.The last four figures in the merged PDF document do not have figure numbers and legends. Was this an error when merging files?

Reviewer: Meichen Yu

REVIEWER COMMENTS

Reviewer #1 (Remarks to the Author):

Reviewer: In this study, Franzmeier et al. investigate the influence of CSF GAP-43 levels on the interaction between amyloid beta pathology and tau PET accumulation rates in a sample of individuals with and without amyloid pathology from ADNI. The authors find that elevated GAP43 levels strengthen the association between amyloid beta and tau accumulation rates in functionally connected regions that were reconstructed in an independent control sample. Overall, the study is well carried out, using appropriate methodology. However, additional clarifications in text on the methods used, as well as on the authors' choice of approach are needed.

Response: We thank the reviewer for these encouraging remarks!

Reviewer: The model used to assess whether the association between functional connectivity and tau pattern rates depended on centiloid and/or GAP43 as described in ll. 376 is not clear. Please provide model specifications (e.g. formulas) more clearly.

Response: We apologize for not providing a clear enough description of the statistical models used in our study. Following the reviewers' comment, we have now added clarification of dependent and independent variable of the regression models in the statistics section on p.18. Specifically, we wrote that *"we tested whether tau epicenter connectivity was more predictive of tau accumulation patterns at higher centiloid and CSF GAP-43 levels. To this end, we assessed first the effect of centiloid alone (i.e. independent variable) on the regression-derived beta values (i.e. dependent variable) that reflected the association between tau epicenter connectivity and subject-level tau-PET change rate patterns, as well as the centiloid x CSF GAP-43 interaction (i.e. independent variable) on the regression-derived beta values."*

Reviewer: In ll.317 the authors refer to previous publications for the model specification to determine tau accumulation rates. However, the linear mixed model was not specified in referred to publications, or may have been missed by this reviewer. Please make sure to include specifications of all (linear mixed) models employed in the study, including covariates, random effects, covariance structure etc., in the manuscript.

Response: The reviewer is correct and we apologize for having cited the wrong publications to describe the linear-mixed model-based calculation of tau-PET change rates. The correct references have now been added on page 15, in which we have used the exact same approach to determine longitudinal tau-PET change rates in the ADNI dataset.¹⁻³ In addition, we have added a more detailed description of the linear-mixed model equation in the manuscript (i.e. tau-PET SUVRs as dependent variable and time from baseline as the independent variable controlling for random slope and intercept). Note that the models described here have been used by us previously to determine tau-PET change rates.¹⁻³ Similarly, others have used the same approach to model change rates in fluid biomarkers in autosomal dominant AD (i.e. NFL).⁴

Reviewer: Related to above, the authors write in ll.148, ll.182 and ll.366 that they used linear mixed models for the interaction effect of GAP43 and centiloids on annual tau accumulation rates. I suspect that linear regressions were instead used. Please correct in text. If not, what random effects (intercept/slope) were used here, as there are no repeated/clustered measurements?

Response: The reviewer is correct that linear regression has been used, as described in the figure legends. We have corrected the text accordingly.

Reviewer: Why was a 30% connectivity density chosen? How were negative correlations handled? The authors should make the reasoning behind their choices clear in the manuscript.

Response: We used our previously developed approach for connectivity-based modelling of tau spreading, that has established a 30% density threshold for the group-average connectivity matrix, with negative correlations being eliminated from the matrix.^{2,5} We have previously performed an extensive set of analyses, showing that applying different density thresholds yields consistent results when modelling tau spreading and when defining Q1-Q4 ROIs.⁵ We now made clear in the methods section (p.16) that we follow this previously established protocol to maximize the consistency across our studies. In addition, we have added supplementary analyses, reporting the regression-based CSF GAP-43 x centiloid interaction on tau accumulation in Q1-Q4 ROIs across different connectivity thresholds (10-40%), which all yielded consistent results (see supplementary table 2).

Reviewer: The authors created a "connectivity template" in an independent sample of cognitively normal controls. Please provide a sample description.

Response: We have added sample demographics of the sample of cognitively normal controls as supplementary table 1, which is now referenced in the manuscript (p.16)

Reviewer: For longitudinal tau scans, did the authors include only two or more images per individual? This should be made clear in the manuscript.

Response: The reviewer is correct that we included two or more scans per individual. We have now briefly clarified this in the methods section (p.13) and have added the average number of tau-PET scans per group to table 1.

Reviewer: In the description for Figure 2 betas are noted, but t values are displayed. Please correct.

Response: We thank the reviewer for catching this error which we have now corrected across all figure legends.

Reviewer #2 (Remarks to the Author):

Franzmeier et al present their investigation on the effect of CSF GAP-43 levels in Aβ⁺ individuals on the speed and pattern of Tau spread, as measured with PET. The manuscript is well written with clear hypotheses and nicely presented data. I see the main finding as that higher CSF GAP-43 levels are predictive for faster disease progression speed in Aβ⁺ individuals after correcting for other factors as age and CSF ptau levels. This is in line with the hypotheses and previous work on GAP-43, a synaptic biomarker. The authors highlight the relevance of synaptic alterations for AD disease progression. Work on CSF GAP43 and on studying mechanisms of disease progression is timely and of potential significance to better understand and prognosticate AD. References are appropriate. While I appreciate that the paper is so focused, the fact that no other protein markers of (pre)synapse are included can also be a weakness. If markers reflecting different processes would have a differential effect on Tau progression, the case for the synaptic changes as driver for Tau-PET progression would strengthen. Now we cannot exclude that higher GAP-43 levels reflects general higher rates of neuronal loss/degeneration including their synapses. Besides this, I have very limited feedback, because the text, figures etc are very clear. I also think the conclusions are phrased with enough caution. I suggest to include a new paragraph on more alternative hypotheses that could explain the results and what to expect (in future studies) including more markers as a broader discussion.

Response: We thank the reviewer for this comment and have added a section to the discussion on the potential of combining multiple synaptic biomarkers to better understand the pathophysiological cascade of AD beyond amyloid and tau. However, we did try not to speculate too much about what to expect from future studies in terms of findings etc., which would be more suitable for a “perspective paper” or an “editorial”. However, we fully agree that studying synaptic biomarkers other than GAP-43 is an important endeavor and we will certainly perform these analyses once data become available in ADNI or are openly shared by others. Yet no other synaptic biomarker is available in ADNI to date, hence studying the complex role of synaptic changes in AD is not possible. Nevertheless, GAP-43 is well described in terms of its molecular role and therefore a promising biomarker to study in the context of tau spreading. Rather than waiting for ADNI to release further and more broad synaptic biomarker data (which may take years or not happen at all), we believe that our current study may actually motivate to assess additional synaptic markers in ADNI and other studies, which will then facilitate future research on synaptic changes in AD and their role in AD pathophysiology and tau spreading. In addition, we have added an alternative explanation of our results on p.11.

Minor:

1. Do the authors understand why in their sample the GAP43 levels are not significantly increased in the AD patients (as referred in introduction).

Response: While statistically not significant, CSF GAP-43 levels are numerically elevated in the Aβ⁺ subjects compared to Aβ⁻ subjects, in line with previous work.⁶ The absence of

a statistically significant effect is most likely linked to the limited number of subjects (i.e. N=93) compared to previous studies showing significant CSF GAP-43 increases across the AD spectrum in ADNI (e.g. N=786 in Ohrfelt et al., Neurology, 2023).⁶ Following the reviewers' comment, we now briefly mention the absence of an A β -related group difference in CSF GAP-43 to the limitations section of the manuscript (p.12).

2. CSF Ptau181 (Elecsys?) is not mentioned in the methods

Response: We have added the reference for the Elecsys-based assessment of CSF p-tau181 in ADNI to the methods section on p.14.

Reviewer #3 (Remarks to the Author):

This manuscript by Franzmeier et al., identified relationships between CSF GAP-43 data and amyloid-beta related tau spread measured by longitudinal PET scans. This is an interesting paper using reasonable approaches to address timely questions in the Alzheimer's disease (AD) field, which largely indicates experience and competence among the authors. Overall, this study was well-designed, and the manuscript is well written. I have a few concerns on the methods and results interpretation, which may be addressed in a revision.

Major comments:

1. The authors claimed that resting-state fMRI data from an independent sample of 42 cognitively normal (CN) controls were used to generate a connectome template. It was not clear to me which independent sample was studied. Did they use the 39 amyloid negative CN subjects from the ADNI? As the number of subjects (42 vs. 39) do not match, I speculate that they were using the HCP or HCP aging (age matched with the ADNI?) data. If so, please clarify why they didn't use the fMRI data from the ADNI, but from another database (as it was not used for replication purpose). How many out of the 93 ADNI subjects have fMRI data? To simulate personalized tau spread, isn't it optimal to use fMRI data from the same subjects of tau PET data? Please also provide an explanation for why data from CN individuals were preferred over patient data in the simulations.

Response: We would like to clarify that we did use subjects from the ADNI database to determine the resting-state fMRI-based connectivity template across the 200 Schaefer ROIs to model connectivity-mediated tau spreading following our previously established protocol^{3,7}. These 42 cognitively subjects were selected independently of the GAP-43 sample (N=93) based on availability of high-quality resting-state fMRI (i.e. 3T, SIEMENS scanner, average framewise displacement < 0.3mm to minimize motion artifacts) a negative amyloid-PET (i.e. global Florbetaben SUVR < 1.08; global Florbetapir SUVR < 1.11) and a negative tau-PET (i.e. global Flortaucipir SUVR < 1.30) scan. A table summarizing demographics and clinical characteristics of this independent cohort was requested by Reviewer 1 and can be found in supplementary table 1. We specifically selected cognitively normal subjects without evidence of neurodegenerative amyloid and tau pathology to ensure that the connectivity template approximates a "normal" connectome in an aging population. Amyloid and tau deposition both have been shown to have an influence on functional connectivity⁸⁻¹¹, which may bias the modelling of connectivity-based tau spreading. Also, the quality of subject-level resting-state fMRI data may vary across sites and subjects and therefore introduce additional bias. Thus, we preferred to use a connectome template derived from a normal control sample to model tau spreading, in line with many of our previous studies^{2,3,7}.

2. On page 15, the authors said that "All individual matrices were averaged and thresholded at 30% density". Was group averaging performed before thresholding? Were there any links with negative correlation values survived after thresholding? If so, how would we estimate/interpret path length with 'negative' correlations (please see PMID:

19819337)? In other words, how would we interpret tau spread via 'negative' links? In addition, were the results sensitive to the choice of link density? Perhaps this can only be answered by validating the results use a range of link density.

Response: We are happy to further clarify our approach of generating the group-average connectivity template. We first averaged the connectivity matrices across subjects, then we eliminated all negative connections followed by applying a density threshold retaining only 30% of the strongest connections. Therefore, no negative connections go into the computation of connectivity-based distance or in the modelling of tau spreading. We have already investigated the effect of applying different density thresholds (i.e. ranging between 10-50%) on the connectivity-based modelling of tau spreading in a previous study, yielding consistent results across thresholds.⁵ To address the influence of connectivity thresholds in the current study, we repeated the entire analysis using thresholds of 10%, 20%, 30% and 40%. Results remained fully consistent across different thresholds as summarized in supplementary table 2. We do not have a specific hypothesis about negative connections and tau spread across these negative connections. Negative or anticorrelated connections can be strengthened by different preprocessing strategies (e.g. global signal regression) and their biological meaning is not entirely clear.¹² Thus, we refrained from including negative connections.

3.The authors used the 200-ROI Schaefer atlas to parcellate the PET and fMRI data. However, in line 116, they said that "In a first step, we tested whether higher A β was associated with faster tau accumulation in predefined summary ROIs that are typically used to quantify tau-PET²⁴". I understand that the authors aimed to use functional connectivity to simulate tau spread, so a functional connectivity based parcellation was used for both the PET and fMRI data. However, the study of reference #24 used FreeSurfer's Desikan-Killiany (D-K) atlas (68 ROIs) to parcellate the PET data. So, I was wondering how the authors used 200-ROI Schaefer atlas to match with the 68 D-K ROIs and created the predefined summary ROIs. Please list the names and coordinates of predefined summary ROIs in a table for replication purpose.

a

Response: The reviewer is correct that the pre-defined summary ROIs were initially defined based on the Desikan-Killiany atlas. However, we have previously mapped the Schaefer atlas to Braak-stage ROIs from the Desikan-Killiany atlas to determine meta regions based on the Schaefer atlas (see Franzmeier et al., Science Advances, 2020).⁵ We have added a spreadsheet of our Schaefer to Braak-stage ROI mapping to the supplementary to facilitate replication, as referenced in the revised version of the manuscript (see Schaefer200_7Networks_Braak_stage_Mapping.xlsx)

4.For some analyses (lines 114 to 126; Fig. 2), it was not clear if both CN and patients' data, or only patients' data were used. Do the colors in Fig. 2 represent groups? Please clarify it.

Response: To avoid any misunderstanding, we have replaced the term "patient-specific" with "subject-specific" and now explicitly mention in the figure legends when the entire

study cohort of 93 subjects is displayed or when only Ab+ subjects are displayed (e.g. Fig.4A). Colors in Figure 2 represent GAP-43 levels as determined via median split for illustrational purposes (see color legend in the figures).

5.This question is related to the last one. Did amyloid-beta negative CN subjects also show similar relationship between CSF GAP-43 data and tau spread as amyloid-beta positive CN and AD patients? Was this relationship stronger in amyloid-beta positive CN than that in amyloid-beta negative CN? If not, shall we think the identified relationships is specific to AD or shared with other tauopathies (e.g., PART or FTD)?

Response: This is an interesting comment, referring to the role of GAP-43 with tau spreading in non-AD tauopathies. We would like to highlight, however, that our main statistical approach investigates the interaction between centiloid (i.e. severity of amyloid) and GAP-43 on tau accumulation and spread, where the effect of GAP-43 was most prominent at higher centiloid levels (e.g. Figs2&3). We did not test a “main effect” of GAP-43 on tau spreading, since the analyses and hypotheses are clearly embedded in an amyloid context following the hypothesis that amyloid causes neurons and synapses to become hyperactive, ensuing GAP43 increases and trans-synaptic tau spread. In amyloid negative subjects, the range of centiloid values is by definition very low, and it is not entirely clear whether centiloid values around -10 to 10 actually pick up minor amyloid deposits or just reflect unspecific binding. Nevertheless, we followed the reviewers suggestion and tested the centiloid x GAP-43 interaction in amyloid negative cognitively normal subjects, showing an overall similar trend as the analyses conducted the amyloid positive subjects (centiloid x GAP-43 interaction on global tau-PET change rates, Controls Amyloid-negative: $T=1.892$, $p=0.068$; All amyloid-positive: $T=2.038$, $p=0.0475$). This suggests that the effect of GAP-43 on amyloid-related tau accumulation already emerges at subthreshold amyloid levels. Since these analyses are in our opinion beyond the focus of the current study, we did not add them to the main manuscript, but they will be accessible to interested reader in the online rebuttal in case the manuscript is accepted. Whether GAP-43 also plays a role in non-AD tauopathies is an interesting question that can, however, not be reliably addressed within the current dataset, since tau imaging in non-AD tauopathies is not established with Flortaucipir¹³. Please note that we conduct another line of research on tau spreading in primary tauopathies (e.g. Franzmeier et al., Nat Commun, 2022 <https://doi.org/10.1038/s41467-022-28896-3>), but analyses on synaptic changes have not been performed in these datasets.

Minor comments:

1.There are some typos in the manuscript. For example, in the Abstract, “and that synapses could be ‘key key’ targets for preventing tau spreading in AD”. One page 5, “To this end, we ‘used linear regression models’ and computed the CSF GAP-43 x centiloid interaction on annual tau-PET change rates using global tau-PET and the temporal meta-ROI as pre-defined readouts ‘using linear regression models’.” Please correct them.

Response: Thanks, we have corrected the typos

2.The last four figures in the merged PDF document do not have figure numbers and legends. Was this an error when merging files?

Response: We guess that Figure numbers are not added in the merged document, which is automatically generated by the submission system. Figure numbers should be, however, included in the main manuscript document. Apologies for the inconvenience.

Reviewer: Meichen Yu

REFERENCES:

- 1 Franzmeier, N. *et al.* The BIN1 rs744373 Alzheimer's disease risk SNP is associated with faster Aβ-associated tau accumulation and cognitive decline. *Alzheimers Dement*, doi:10.1002/alz.12371 (2021).
- 2 Frontzkowski, L. *et al.* Earlier Alzheimer's disease onset is associated with tau pathology in brain hub regions and facilitated tau spreading. *Nat Commun* **13**, 4899, doi:10.1038/s41467-022-32592-7 (2022).
- 3 Pichet Binette, A. *et al.* Amyloid-associated increases in soluble tau relate to tau aggregation rates and cognitive decline in early Alzheimer's disease. *Nat Commun* **13**, 6635, doi:10.1038/s41467-022-34129-4 (2022).
- 4 Preische, O. *et al.* Serum neurofilament dynamics predicts neurodegeneration and clinical progression in presymptomatic Alzheimer's disease. *Nat Med* **25**, 277-283, doi:10.1038/s41591-018-0304-3 (2019).
- 5 Franzmeier, N. *et al.* Patient-centered connectivity-based prediction of tau pathology spread in Alzheimer's disease. *Sci Adv* **6**, doi:10.1126/sciadv.abd1327 (2020).
- 6 Ohrfelt, A. *et al.* Association of CSF GAP-43 With the Rate of Cognitive Decline and Progression to Dementia in Amyloid-Positive Individuals. *Neurology* **100**, e275-e285, doi:10.1212/WNL.0000000000201417 (2023).
- 7 Franzmeier, N. *et al.* Patient-centered connectivity-based prediction of tau pathology spread in Alzheimer's disease. *Sci Adv* (2020).
- 8 Schultz, A. P. *et al.* Phases of Hyperconnectivity and Hypoconnectivity in the Default Mode and Salience Networks Track with Amyloid and Tau in Clinically Normal Individuals. *J Neurosci* **37**, 4323-4331, doi:10.1523/JNEUROSCI.3263-16.2017 (2017).
- 9 Huijbers, W. *et al.* Amyloid-beta deposition in mild cognitive impairment is associated with increased hippocampal activity, atrophy and clinical progression. *Brain* **138**, 1023-1035, doi:10.1093/brain/awv007 (2015).
- 10 Huijbers, W. *et al.* Tau Accumulation in Clinically Normal Older Adults Is Associated with Hippocampal Hyperactivity. *J Neurosci* **39**, 548-556, doi:10.1523/JNEUROSCI.1397-18.2018 (2019).
- 11 Guzman-Velez, E. *et al.* Amyloid-beta and tau pathologies relate to distinctive brain dysconnectomics in preclinical autosomal-dominant Alzheimer's disease. *Proc Natl Acad Sci U S A* **119**, e2113641119, doi:10.1073/pnas.2113641119 (2022).
- 12 Murphy, K. & Fox, M. D. Towards a consensus regarding global signal regression for resting state functional connectivity MRI. *Neuroimage* **154**, 169-173, doi:10.1016/j.neuroimage.2016.11.052 (2017).
- 13 Lemoine, L., Leuzy, A., Chiotis, K., Rodriguez-Vieitez, E. & Nordberg, A. Tau positron emission tomography imaging in tauopathies: The added hurdle of off-target binding. *Alzheimers Dement (Amst)* **10**, 232-236, doi:10.1016/j.dadm.2018.01.007 (2018).

REVIEWER COMMENTS

Reviewer #1 (Remarks to the Author):

- Regarding my first comment on the initial submission, I appreciate that the authors tried to add some more details. However, in the current version of the manuscript, p. 18, ll. 411, still reads convoluted. Specifically, I strongly suspect readers will have trouble following from which model beta values were derived that were used as outcome for this model. It would greatly improve readability to specify all models in formulas as per initial suggestion. Alternatively, the authors should expand this paragraph to guide the readers through their analysis.
- Please correct the typos in Supplementary Table 1 for the mean age of the sample (671.7 years) and table 1 for the number of tau-PET visits for CN Abeta negative 2.79 ± 1.13 .

The authors have addressed my other comments.

Reviewer #2 (Remarks to the Author):

To be honest, while I had only few comments the authors have not really addressed my main comment. I can understand that it is not feasible to include new markers in the analysis, but the interpretation of the results is quite broad, on a biological level. I was especially surprised with: 'Yet no other synaptic biomarker is available in ADNI to date, hence studying the complex role of synaptic changes in AD is not possible.'

A brief search on synaptic markers SNAP-25 and neurogranin, neuropentaxin:

<https://doi.org/10.1186/s13195-018-0407-6>,

https://adni.bitbucket.io/reference/docs/FAGANLAB/ADNI_Methods_Fagan_Lab_Final.pdf, and [10.1002/alz.12353](https://doi.org/10.1002/alz.12353).

Reviewer #3 (Remarks to the Author):

The authors have addressed all my concerns. I don't have further comments. Great work!

Meichen Yu

Reviewer #1 (Remarks to the Author):

Reviewer:- Regarding my first comment on the initial submission, I appreciate that the authors tried to add some more details. However, in the current version of the manuscript, p. 18, ll. 411, still reads convoluted. Specifically, I strongly suspect readers will have trouble following from which model beta values were derived that were used as outcome for this model. It would greatly improve readability to specify all models in formulas as per initial suggestion. Alternatively, the authors should expand this paragraph to guide the readers through their analysis.

- Please correct the typos in Supplementary Table 1 for the mean age of the sample (671.7 years) and table 1 for the number of tau-PET visits for CN Abeta negative 2.79 ± 1.13 .

Response: We thank the reviewer again for this comment and now explicitly mention all linear model equations in the statistics section of the manuscript (p.18). In addition, we have corrected the typos in supplementary table 1 and table 1.

Reviewer: The authors have addressed my other comments.

Response: Thanks for these encouraging remarks!

Reviewer #2 (Remarks to the Author):

To be honest, while I had only few comments the authors have not really addressed my main comment. I can understand that it is not feasible to include new markers in the analysis, but the interpretation of the results is quite broad, on a biological level. I was especially surprised with: 'Yet no other synaptic biomarker is available in ADNI to date, hence studying the complex role of synaptic changes in AD is not possible.'

A brief search on synaptic markers SNAP-25 and neurogranin, neuropentaxin:

[https://doi.org/10.1186/s13195-018-0407-](https://doi.org/10.1186/s13195-018-0407-6)

[6,https://adni.bitbucket.io/reference/docs/FAGANLAB/ADNI_Methods_Fagan](https://adni.bitbucket.io/reference/docs/FAGANLAB/ADNI_Methods_Fagan) Lab Final.pdf, and 10.1002/alz.12353.

Response:

We would like to thank the reviewer again for his comments and remarks, which we are happy to discuss in more detail: First, we appreciate the reviewers' suggestion to include other markers and pointing us to potentially suitable datasets. We were and are indeed aware of the existing CSF measures from the Fagan lab as well as CSF neurogranin measures from the Blennow lab that have been uploaded to the ADNI database in the past. While including these data would be in principle of great value to address the role of synaptic biomarker changes in tau spreading, the Fagan lab CSF assessments cover years 2005-2013 and the Blennow Lab the years 2005-2007, which is long before tau-PET was introduced to ADNI in late 2015. We carefully cross-matched again the Blennow/Fagan lab data referenced above with our current sample, and there is indeed no subject that meets our inclusion criteria (i.e. baseline amyloid-PET and longitudinal tau-PET) and has other synaptic biomarkers available. Thus, the currently available ADNI data do not allow studying other synaptic biomarkers in the role of tau spreading assessed with longitudinal Flortaucipir tau-PET. Going back to actively obtain new synaptic markers would clearly go beyond the scope of the current study, as also acknowledged by the reviewer. We agree, however, that our statement referenced by the reviewer may be misleading and imply that no synaptic biomarker data apart from GAP-43 are available in ADNI at all. We have therefore changed our wording to "*Also, no other synaptic biomarker beyond GAP-43 is currently available in close enough proximity to longitudinal tau-PET assessments in open access datasets such as ADNI, which clearly limits the study of synaptic changes in tau spreading in AD.*" (p.12). This modified statement should specifically account for the missing overlap of tau-PET

and other synaptic biomarkers and illustrate why none of the other markers could be included in the current study. We hope that this modification addresses the reviewers' concern and clarifies the potential availability of other synaptic markers in ADNI that might be helpful to address other questions on synaptic changes in AD.

Second, the reviewer mentions that the interpretation of our results is quite broad, on a biological level. We would like to highlight that the reviewer has previously commented that our paper is very focused, potentially too focused, and has therefore asked us *"to include a new paragraph on more alternative hypotheses that could explain the results and what to expect (in future studies) including more markers as a broader discussion."* Following the reviewers' initial assessment during the previous round of revisions, we have therefore added a specific section to the discussion on the importance of combining multiple synaptic biomarkers in future studies and have discussed alternative hypotheses that may explain our results (see p.11, *"As an alternative explanation, GAP-43 and p-tau may mirror higher Ab-related neuronal activity and metabolism, which may lead to overall higher transcriptional and translational activity of GAP-43 and tau, ensuing faster local tau aggregation independent of trans-synaptic tau spread"*). Thus, we primarily included a broader discussion because the reviewer asked us to do so, hence we interpret the current comment as slightly contradictory to the initial assessment and we apologize if we have somehow misunderstood the reviewer during the initial review. We appreciate that a broader evidence-based discussion of other synaptic biomarkers would be desirable, but we argue again that an evidence-based scientific discussion would also require actual analyses and results. As we outlined above, no other synaptic biomarkers beyond GAP-43 can be included in the current study, hence we still refrain from actively discussing the role of other synaptic proteins in tau spreading, which would be speculative in the absence of data. However, once more data become available in ADNI or from our collaborators, we will definitely follow this line of research further and hope to address the reviewers' wish for a more complex role of the "synaptome" in tau spreading in the future.

Reviewer #3 (Remarks to the Author):

Reviewer: The authors have addressed all my concerns. I don't have further comments. Great work!

Meichen Yu

Response: We thank the reviewer for the helpful and encouraging comments and supporting us in improving the paper!